# Pediatric Primary Care Perspectives of Mental Health Services Delivery during the COVID-19 Pandemic

**DOI:** 10.3390/children9081167

**Published:** 2022-08-03

**Authors:** Chuan Mei Lee, Jessica Lutz, Allyson Khau, Brendon Lin, Nathan Phillip, Sara Ackerman, Petra Steinbuchel, Christina Mangurian

**Affiliations:** 1Department of Psychiatry and Behavioral Sciences, Weill Institute for Neurosciences, University of California San Francisco, San Francisco, CA 94143, USA; christina.mangurian@ucsf.edu; 2Clinical Excellence Research Center, School of Medicine, Stanford University, 453 Quarry Road, Palo Alto, CA 94304, USA; 3Benioff Children’s Hospitals Child and Adolescent Psychiatry Portal, University of California San Francisco, 5100 Martin Luther King Jr. Way, Oakland, CA 94609, USA; jlutz@berkeley.edu (J.L.); allyson.khau@sjsu.edu (A.K.); blin6@pride.hofstra.edu (B.L.); nathanphillip39@gmail.com (N.P.); petra.steinbuchel@ucsf.edu (P.S.); 4Department of Social and Behavioral Sciences, University of California San Francisco, 490 Illinois St., Floor 12, Box 0612, San Francisco, CA 94143, USA; sara.ackerman@ucsf.edu; 5Department of Epidemiology and Biostatistics, University of California San Francisco, 550 16th St., 2nd Floor, San Francisco, CA 94158, USA; 6Center for Vulnerable Populations, Zuckerberg San Francisco General Hospital, 2789 25th St., San Francisco, CA 94110, USA; 7Philip R. Lee Institute for Health Policy Studies, University of California San Francisco, 490 Illinois St., San Francisco, CA 94158, USA

**Keywords:** COVID-19 pandemic, qualitative, pediatric primary care, mental health services

## Abstract

Due to a national shortage of child and adolescent psychiatrists, pediatric primary care providers (PCPs) are often responsible for the screening, evaluation, and treatment of mental health disorders. COVID-19 pandemic stay-at-home orders decreased access to mental health care and increased behavioral and emotional difficulties in children and adolescents. Despite increased demand upon clinicians, little is known about mental health care delivery in the pediatric primary care setting during the pandemic. This focus group study explored the experiences of pediatric PCPs and clinical staff delivering mental health care during the pandemic. Transcripts from nine focus groups with San Francisco Bay Area primary care practices between April and August 2020 were analyzed using a thematic analysis approach. Providers expressed challenges at the patient-, provider-, and system-levels. Many providers reported increased patient mental health symptomatology during the pandemic, which was often intertwined with patients’ social determinants of health. Clinicians discussed the burden of the pandemic their own wellness, and how the rapid shift to telehealth primary care and mental health services seemed to hinder the availability and effectiveness of many resources. The findings from this study can inform the creation of new supports for PCPs and clinical staff providing mental health care.

## 1. Introduction

The novel coronavirus (COVID-19) public health crisis and its associated social and economic repercussions have exacerbated problems of access to mental health care for youth [1]. High rates of depression, anxiety, and overall worse mental health have been reported among children and adolescents since the onset of the COVID-19 pandemic, increasing demand for mental health services [2,3]. However, even before the pandemic, there have been significant mental health access issues in the setting of a severe national shortage of child and adolescent psychiatrists and other mental health professionals [4,5]. The communities most critically affected by this workforce shortage are rural, low-income, and racial/ethnic minoritized youth—often served in safety-net settings like federally qualified health centers (FQHC) [6,7]. These communities are at higher risk of mental health challenges during the COVID-19 pandemic [8].

Prior to the COVID-19 pandemic, one study found that over one-third (34.8%) of youth between the ages of 2–21 received their outpatient mental health care from a primary care provider (PCP) only [9]. Given the increased risk of developing mental health symptoms and decreased access to mental health services during the pandemic (with reductions in school-based mental health) [1], pediatric PCPs and clinical staff (e.g., nurses, social workers, and integrated-care psychologists) have become critical for evaluating and treating youth with mental health concerns. Despite this, many PCPs have long expressed that they often do not feel comfortable treating or managing mental health conditions, or do not feel they have adequate time to address mental health problems in primary care [10,11]. Therefore, there is growing need for psychiatrists and other mental health specialists to support and partner with their primary care colleagues to deliver adequate mental health services during the pandemic and in its aftermath. Promising supports include integrated behavioral health programs, such as child psychiatry access lines, which provide PCPs with real-time telephonic consultation with child psychiatrists [12,13,14].

However, little is known about mental health service delivery in pediatric primary care during the COVID-19 pandemic. This qualitative research study aimed to understand the experiences of pediatric PCPs and clinical staff delivering mental health care during the pandemic through focus group interviews to improve integrated behavioral health programs that support primary care.

## 2. Methods

This focus group study was designed to understand mental health care delivery experiences among pediatric PCPs and clinical staff. This study was nested within a larger effort to implement the UCSF Child and Adolescent Psychiatry Portal (CAPP) program, modeled after other psychiatric access programs [12,13], which provides free telephonic pediatric psychiatry consultation to PCPs throughout California. The Consolidated Criteria for Reporting Qualitative Research (COREQ) Checklist was used to guide this study [15].

### 2.1. Recruitment

We recruited a convenience sample of community clinics and FQHCs in the San Francisco Bay Area that were participating in the initial launch of the UCSF CAPP program April to August 2020. We emailed the clinic manager to solicit clinician participation in a focus group at the time of UCSF CAPP enrollment. Focus groups were recruited until data saturation was achieved.

The main inclusion criterion for participants from these clinics was the provision of pediatric clinical services at the clinic. We included pediatricians, nurse practitioners, an advice nurse, family medicine physicians, social workers, psychologists, and a community health worker. Non-direct patient care staff members, such as administrative staff, were excluded. All participants completed informed consent and received a $50 gift card for their participation. Ethical approval (IRB number 2019-117) was obtained from the UCSF Benioff Children’s Hospital Oakland institutional review board.

### 2.2. Study Procedures

Focus groups took place online via secure video conferencing services between April and August 2020. Each focus group consisted of clinicians from the same primary care practice. As this study was nested within the UCSF CAPP enrollment process, participants received an orientation to UCSF CAPP and then participated in the focus group. Focus groups lasted ~45 min and were audio recorded on an encrypted device. A child psychiatrist (CML) trained in qualitative research methods moderated the focus groups, and a notetaker (JL, AK, NP) was present. Additionally, we collected de-identified demographic information on all participants (Table 1).

The semi-structured focus group interview guide (available upon request) for this study was developed by study researchers (CML, PS) based on prior experience providing child psychiatric consultation to primary care providers and was pilot tested at one pediatric primary care clinic in 2019. The guide was developed to understand the practice’s capacity to treat mental health care among child and adolescent patients, referral systems, and facilitators and barriers to mental health care delivery, and it was initially deployed in April 2020. As this period coincided with the start of the COVID-19 pandemic in the United States, additional questions specifically querying provider experiences during the pandemic were added and IRB-approved by June 2020. Of note, the study sample was restricted to those focus groups that discussed the COVID-19 pandemic.

## 3. Data Analysis

Focus group audio recordings were transcribed verbatim and coded independently by two graduate-student researchers (JL, AK) using ATLAS.ti Mac (Version 8.4.4). The researchers utilized a thematic analysis approach, which involved an inductive and iterative process of coding and interpretation to derive themes [16]. A preliminary codebook was created by several members (JL, AK, BL, NP) of the research team. Two researchers (JL, AK) independently applied the codebook to the transcripts, then compared their coded transcripts, and consolidated them into a single ATLAS.ti file. Any differences were resolved through discussion and consensus. Further discussions identified key themes related to the COVID-19 pandemic. We then organized these themes into domains by applying an existing social-ecological framework [17], which posits that individual, relationship, community, and societal factors converge and interact to influence health. We chose this framework because the social-ecological model recognizes that an individual child does not exist in isolation but is nested within family and clinician relationships, community contexts, and social policies. We represented participant quotes by clinic number.

## 4. Results

Overall, we conducted 11 focus groups. However, we excluded two focus groups from the analysis because these groups did not receive the COVID-19-specific questions and did not discuss experiences pertaining to COVID-19. Therefore, we analyzed comments from a total of nine focus groups. Each focus group consisted of two to ten providers from a pediatric primary care clinic across five San Francisco Bay Area counties: Alameda, Contra Costa, Marin, San Francisco, and San Mateo. Upon recruitment, 50 providers indicated their intent to participate. A total of 48 providers completed the focus group interview. Among those providers, 54% (26/48) worked in FQHCs, and 46% (22/48) worked in community practices (Table 1).

We identified five main themes derived from the data: (1) rapid transition to virtual appointments, (2) limited availability of behavioral health resources, (3) impact of the pandemic on providers, (4) changes in mental health symptomatology, and (5) social determinants of health (See Table 2 for additional quotes). Applying a social-ecological framework, we organized themes into three domains: system-, provider-, and patient-level (Figure 1). In this context, the individual experiences of the patients and providers, as well as their interpersonal interactions, occur within the context of the mental health care system and society. These simultaneous, evolving interactions sum to influence a provider’s experience providing mental health care during the COVID-19 pandemic.

## 5. System-Level Domain

### 5.1. Rapid Transition to Virtual Appointments

As clinics transitioned to virtual appointments during the pandemic, providers across focus groups described virtual appointments as both a facilitator for and barrier to care. Providers noted that virtual appointments were convenient for patients due to reduced cost and travel time, and increased accessibility for people living with physical disabilities. One provider wondered whether the “*anonymity*” of virtual appointments could reduce fears of stigma associated with seeking behavioral health care (Clinic 8). However, loss of the patient-provider connection was a frequent concern as providers reported challenges engaging with patients, especially young children, and building relationships with new patients via telehealth.

Another consequence of the shift to virtual appointments was screening challenges. Providers reported barriers to screening for mental illness, substance use, and adverse childhood experiences because most screenings were formerly conducted via a self-report questionnaire in the waiting room. For example, a provider reported that the lack of routine wellness appointments during the pandemic provided fewer opportunities to screen patients for mental health concerns and connect them to resources (Clinic 6).

Another concern was confidentiality. Several providers reported concerns about patients’ access to private spaces, particularly for those living in shared family homes, which they believe hindered patients’ ability to be honest and vulnerable during virtual appointments: “*It’s two families living in a single bedroom home or the whole family is in a room. So at least when they come into the clinic, we have the opportunity to kick the parent out, ask really private questions*” (Clinic 5).

### 5.2. Limited Availability of Behavioral Health Resources

Clinic sites with integrated behavioral health services or clinic coordinators that help schedule mental health visits reported new challenges with their warm hand-off models during the pandemic. Prior to the pandemic, warm hand-off models typically involved an in-person introduction to a referral coordinator during the visit. During the pandemic, providers reported that separate phone calls to the patient from the physician and coordinator had disintegrated the collaboration inherent to the warm hand-off model. Ultimately, this contributed to a disconnect between routine health services and mental health services.

Providers’ experiences with mental health referral processes varied. Some providers reported that they had not noticed changes to their mental health referral process, while others reported difficulties because of mental health clinics being closed or overwhelmed during the pandemic. Providers suggested that the capacity of these referrals was also limited by a lack of staffing: office staff were more frequently taking sick leave or time off to care for family members. Among those who commented that they had not noticed a change in the referral process, one provider suggested that it was too soon to notice (August 2020) because prior to the pandemic there was already a lengthy waiting period.

Several providers commented on the effect of the pandemic on school-based resources: “*Parents are really stressed… and not getting as much support from the schools*” (Clinic 1). Although some school-based resources remained available, many students who previously relied on these resources for learning disabilities or other wellness services lost access during the pandemic due to school closures.

## 6. Provider-Level Domain

### Impact of the Pandemic on Providers

Providers discussed the increased stress that they experienced working in the medical setting during the pandemic. Some spoke explicitly about the fear of dying from COVID-19: “*As clinicians, you have your own fear… I don’t want to catch it. I don’t want to die from COVID*” (Clinic 3). Providers from community clinics shared the financial stresses of running a healthcare business during a pandemic: “*We count ourselves fortunate that we’re still open. I know lot of other practices are not, and so there is that stress. It’s a tough time*” (Clinic 3). Some providers also discussed the emotional challenges of being a “*de facto trusted reference*” for patient families, given the lack of a unified national message in the first few months of the pandemic (Clinic 3).

Providers creatively adapted services to meet patients’ needs during the pandemic, sometimes dramatically changing their workflow and workload. Some examples included shifting clinic services to remote formats, having weekly telehealth check-ins with patients who were unable to see their therapist, and creating internal processes to offer direct counseling to households in which a family member tested positive for COVID-19. Another stated impact on providers was the need for more time to interact with patients during check-ups, video visits, and phone calls to address the impact of COVID-19 on patients’ lives and health.

## 7. Patient-Level Domain

### 7.1. Changes in Mental Health Symptomatology

Providers observed increases in mental health symptoms among patients, especially anxiety, depression, and loneliness. For example, a clinician noted, “*We’ve done some surveying of community needs after COVID hit and shelter-in-place, [which] revealed a heightened level of anxiety and recurring trauma, depressive feelings, sense of chaos and shortage in the world, and just in general a heightened need*” (Clinic 8). Providers attributed some of these symptoms to social isolation and/or increased time at home with family. Providers described a cascade of stress from parents to children during the pandemic, highlighting how community disparities “*trickle down*” to affect kids (Clinic 7).

Somatic symptoms were also commonly observed, including upticks in stomachaches, worsening diabetes, and worsening hypertension. Providers suspected that in some cases these symptoms could be attributed to underlying behavioral health concerns: “*I feel like I’m seeing somatization if that’s the right word. I mean the number of kids in the past two and a half weeks who have had stomach aches or accidents… is probably more than the previous three months put together*” (Clinic 3).

Providers recalled parental/guardian concerns encountered during the pandemic, including video gaming addiction, running away from home, or suicidal ideation: “*Three-year-olds, four-year-olds, and five-year-olds, are really starting to act out and say kind of troubling things about wanting to die and wanting to kill themselves*” (Clinic 4).

Conversely, many providers also reported that school closures during the pandemic had positive effects on mental health for some of their patients. They observed that students affected by social anxiety, bullying, and academic challenges experienced relief from school-based stressors while sheltering-in-place: “*A lot of the kids who are suffering from some social anxiety because they were going to school, all of a sudden they felt great*” (Clinic 5).

In parallel to the overall increased mental and behavioral health symptoms observed by providers, two practices discussed the increased utilization of behavioral health services among patients. One practice reported a 25% increase in utilization of behavioral health services across three clinic sites, and another noted an increase in therapy needs.

### 7.2. Social Determinants of Health

Many clinicians discussed the impact of families’ socioeconomic status (SES) on pediatric mental health during the pandemic. A clinician in a community practice primarily serving patients with private insurance observed that their patients’ demographics allowed them to access outdoor yard space at home, which helped them cope with isolation: “*I feel like a lot of our families and their kids seem to be coping really well and that might be kind of our demographics. Where—it’s easier. They have houses with yards and some of them are still having help come in*” (Clinic 2).

Providers in FQHC practices discussed the economic hardships faced by their patients’ families during the pandemic. Some reported an increase in patients due to families losing employment and corresponding insurance. Additionally, FQHC providers discussed socioeconomic barriers to virtual care, such inability to pay the phone bill or lacking access to technology to access healthcare appointments. They described the necessity of considering a patient family’s SES when determining a course of mental health care. Another provider noted that the civil unrest against racial injustice following the murder of George Floyd in the summer of 2020 compounded the burdens and uncertainty of the pandemic.

## 8. Discussion

This study revealed experiences of pediatric primary care clinicians delivering mental health care in primary care settings during the first year of the COVID-pandemic. Overall, healthcare services changed fundamentally at the system level, given the rapid shift to virtual appointments for mental health care, which resulted in both facilitators and barriers to care. Providers stated that virtual appointments facilitated medical visits because they reduced the need for patient and family travel to the clinic. However, PCPs also reported patients’ difficulties with accessing technology (devices, Internet, etc.), computer literacy, and obtaining private areas for telehealth visits—all of which they attributed to low SES. Other clinical literature has noted the financial burden of telehealth on families [18], and this speaks to the need to partner with government or community agencies that may provide technology resources. If a PCP is sensing that a patient is lacking privacy at home, then that patient may need to be prioritized for an in-person visit. Providers also acknowledged difficulty in engaging with pediatric patients, particularly young children, via telehealth, which has been cited in the literature as a significant barrier to remote mental health treatment [19,20]. However, clinicians may work on engaging the adult caregiver as an ally in establishing a therapeutic alliance with the child, for example asking the adult to facilitate play with the patient. Nonetheless, as the pandemic progresses, a hybrid care model incorporating options for both virtual and in-person mental health appointments should be considered.

Overall, the pandemic exacerbated pre-existing challenges to the provision of mental health care in primary care. While there was some variation in experience, many providers reported perceiving a reduction in available behavioral health resources. Providers reported long wait times for patient mental health referrals that existed well before the pandemic. The significant challenge in referring patients to specialty mental health care due to the limited mental health workforce could be mitigated by integrated and consultative care models, such as the Collaborative Care Model or Child Psychiatry Access Programs, which help to extend the reach of child mental health specialists [21,22].

Providers themselves also reported experiencing new stressors during the pandemic, including personal fears of COVID-19 infection and increased workloads from adapting services to the pandemic. Clinicians noted that there was the need for more time to interact with patients during telehealth encounters to address the impact of COVID-19 on patients’ lives. Because the focus groups were conducted early in the pandemic, this finding may be attributable to an initial adjustment to the pandemic. However, considering increased and ongoing stress from multiple sources during the pandemic, assessing and supporting healthcare worker well-being is critical.

At the patient level, providers observed increased patient symptomatology and socioeconomic stress. In many cases, our findings in this study aligned with existing literature suggesting that the pandemic has had an overall negative effect on child and adolescent mental health [1,2,23]. While some providers in our study noticed that children with social anxiety, bullying, and academic challenges initially experienced relief during school closures, the literature supports that lockdown conditions prevented exposure to feared situations and reinforced avoidant behaviors, likely making existing mental health conditions worse when schools reopened [24]. Providers described how patients’ pandemic living situations (e.g., overcrowding, lack of private spaces) affected mental health. Providers also discussed elevated stress among families with essential workers due to increased risk for infection or loss of employment. The provision of mental health care requires awareness of patients’ SES and resources both during and after the COVID-19 pandemic.

## 9. Limitations

This study represents a subset of providers in one metropolitan area and may not be generalizable to all providers, particularly those practicing in rural settings. Nonetheless, our study included a diverse group of providers across the San Francisco Bay Area. Additionally, the experiences of providers from different delivery settings were included, reflecting a socioeconomically diverse range of patient populations.

The timeline of our study allowed us to observe the impact of the rapidly changing pandemic at different points in time; however, this hindered comparisons between practices. Specifically, FQHC and community practice providers were interviewed at different times during the pandemic. Focus groups with community practices took place between April and August 2020, while focus groups with FQHCs took place in August 2020. Provider quotes were obtained at one point in time and may not represent changing perceptions throughout the pandemic.

## 10. Conclusions

Our study highlights the longstanding and increasing mental health needs of children and adolescents during the COVID-19 pandemic, as well as the increasing complexity of mental health problems that PCPs regularly face in primary care. PCPs have highlighted the increasing challenges to their workloads and their own wellbeing. As the pandemic continues to impact our most vulnerable communities, the findings from this study can inform new collaborations between specialty mental health and primary care to support PCPs and clinical staff delivering mental health care.

## Figures and Tables

**Figure 1 children-09-01167-f001:**
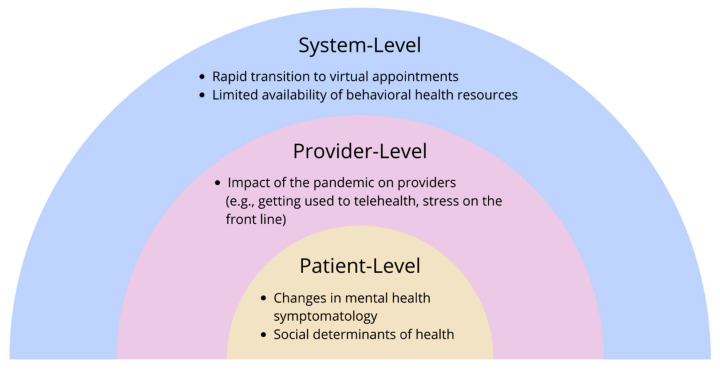
Factors influencing pediatric primary care providers’ ability to deliver mental health care during the COVID-19 pandemic as reported by participants, an adapted social-ecological framework.

**Table 1 children-09-01167-t001:** Participant demographics.

Provider Demographics	(N = 50) ^1^	
	N	%
Age		
18–29	2	4.0
30–39	19	38.0
40–49	15	30.0
50–59	10	20.0
60–69	4	8.0
Race/ethnicity		
Asian	17	34.0
Black or African American	2	4.0
Hispanic/Latinx	4	8.0
White	21	42.0
Mixed Race	4	8.0
Other	1	2.0
Decline to state	1	2.0
Gender		
Female	43	86.0
Male	7	14.0
Provider type		
Community Health Worker	1	2.1
MD (pediatrics)	28	58.3
MD (family)	5	10.4
Medical Resident	1	2.1
NP (general)	3	6.3
NP (pediatrics)	3	6.3
NP (psych)	1	2.1
PhD	1	2.1
PsyD	1	2.1
RN	1	2.1
Social Worker	3	6.3
Years of practice/work experience		
0–5	14	28.0
6–10	8	16.0
11–15	12	24.0
16–20	5	10.0
21+	11	22.0
Patients’ predominant insurance type		
Commercial insurance	23	46.0
Medi-Cal	21	42.0
Other	4	8.0
Unknown	2	4.0
Practice Type		
FQHC	26	54.2%
Community Practice	22	45.8%
Provider language(s) spoken other than English		
None	19	38.0
Spanish	18	36.0
Hindi	5	10.0
Japanese	3	6.0
Mandarin	3	6.0
French	2	4.0
German	2	4.0
Urdu	2	4.0
Arabic	1	2.0
Hebrew	1	2.0
Kannada	1	2.0
Punjabi	1	2.0
Telugu	1	2.0
Yoruba	1	2.0

^1^ Fifty providers indicated intent to participate, however only 48 providers completed the study. The two providers who did not participate in focus groups could not be removed from the demographic analysis because this information was collected anonymously. These two providers are not included under “Provider Type”.

**Table 2 children-09-01167-t002:** Domains, Themes, and Representative Quotes from Providers on the Impact of the COVID-19 Pandemic on Mental Health Care Delivery in Pediatric Primary Care.

System-Level Domain	
Theme: Rapid Transition to Virtual Appointments	
Subtheme: Increased Convenience	*“Since almost everything is virtual, they don’t have to get themselves anywhere, they can just hop on a computer and Zoom with somebody.” (Clinic 1)* *“I have also seen some positives in the ability of people to access care through telemedicine, that a family doesn’t have to take time off work and get in the car and drive somewhere and find parking, and it actually reduces the stress of them getting the care that they need. And it makes the access to care somewhat easier in many ways. So lots of negatives, but also some positives…the ability of us to do telemedicine I hope stays.” (Clinic 7)* *“I think in terms of…benefits to be able to have…physical mobility, disabilities, or even related stigma…barriers to walking into a behavioral health building for services. There’s a level of anonymity…that technology affords…as well as thinking in terms of reducing cost and time for the patient, a disability maybe for some of that. So we’ve definitely seen a much lower no-show rate.” (Clinic 8)*
Subtheme: Barriers to Patient-Provider Connection	*“I think our model of care has really changed during COVID I think one of the very unique things of our clinic, it was very much based on an open-door sort of model. Especially for our patient population that people could really stop by. We would tend to see a lot of drop-ins everyday which I think really served our patients. Often it was not necessarily for mental health but it was for something else, but what came up was actually a moment of crisis and needing somebody to talk to, needing somebody to be connected to, and that is just not available anymore. We really have closed our doors. Not in the sense of keeping people out but trying to keep our staff and providers safe, our patients safe, and really only seeing people who need to come into clinic but that’s very hard to figure out. You can’t just walk into clinic anymore. I think that has really changed our ability to connect to patients and connect patients with additional resources.” (Clinic 6)* *“If you are seeing a patient for the first time, or a patient you’re not familiar with yourself as a primary care provider, which can happen quite a bit, especially in COVID. And so you don’t have a good grasp of the social family dynamics, and it seems like the patient has severe symptoms, and you’re wondering, ‘how do I approach this now, and handle this now, before I fall out with them.’” (Clinic 7)* *“Working with younger patients through virtual means has been a challenge for our providers who do work with pediatrics; it’s just harder to engage with a five-year-old through video for therapy services.” (Clinic 8)*
Subtheme: Screening Challenges	*“I find it uncomfortable to go down that ACEs [adverse childhood experiences] checklist by phone, so I don’t do it…for a family that’s new to our clinic…for brand new patients to go down a list of potentially traumatic questions, I find very difficult. I don’t know how they perceive it on their end” (Clinic 5)* *“I also think assessing risk is harder over the phone. Yeah, it’s been scarier for me to do that over the phone.” (Clinic 6)* *“I think the fact that we’re not doing routine visits on teens anymore, as well. Because they don’t need them for school. Right? And because that’s not an urgent service. It means we’re missing the things that came up on routine physicals in terms of mental health screening and sexual health screening stuff. We’re missing opportunities to get people connected to mental health care or get them birth control or get them other sensitive services that were coming up in the context of sort of general preventative care that we’re just not able to do.” (Clinic 6)*
Subtheme: Confidentiality Concerns	*“Especially for adolescents, the confidentiality piece is just so hard because we have a lot of our families live in shared homes and it’s two families living in a single bedroom home or the whole family is in a room. So at least when they come into the clinic, we have the opportunity to kick the parent out, ask really private questions.” (Clinic 5)* *”…that confidentiality piece on the phone—that they might not be willing to be vulnerable when they can’t be or if they—there might be other people around them.” (Clinic 5)*
**Theme: Limited Availability of Behavioral Health Resources**	
Subtheme: Warm Hand-off Model	*“Warm hand off model has sort of disintegrated.” (Clinic 6)* *“Now with COVID that our behavioral health clinicians are working remotely, and our medical providers are onsite, and that we’ve been understaffed for behavioral health for quite a while, that system has broken down a little bit. So, it’s a little more challenging to do warm hand off and to really get that in the moment collaboration that we would like.” (Clinic 6)* *“Our warm hand off model—I think it feels very disconnected for patients that they don’t associate necessarily me with [name] because we’re calling at two different times. So, I think the connection between your routine health care and your mental health services is kind of getting broken apart.” (Clinic 6)*
Subtheme: Referrals	*“All of us don’t feel comfortable once the kid is complicated in family medicine, or suicidal. I don’t think that’s where pediatricians are supposed to be managing patients, and we’re always trying to refer them on. It’s been really really hard to get them [families] to use the system. And then when they do try to use the system, the system doesn’t always work.” (Clinic 1)* *“It does sort of feel like the therapists and the psychiatrists are becoming less available. I think there’s definitely a greater need.” (Clinic 1)* *“I haven’t noticed a difference because it is still—I haven’t noticed it worsen but I don’t think it was very likely that I could have a patient call [a behavioral health access line] and get an appointment any sooner than 3–6 months. I don’t think that’s changed. I don’t think we’ve been in COVID long enough for me to see a change because it was already taking so long.” (Clinic 5)* *“Sometimes somebody is out sick, and the referral doesn’t get processed or on the other side with people receiving the referrals, their front desk is out sick. Just with normal, maybe not with COVID illness, but taking care of their kids. We—a lot of people—now that school is going to start up, moms are calling out sick, taking FMLA. Staffing just in general in the medical system I think is lower. And so, while I haven’t seen it yet, I would predict that wait times do get longer just because of overwhelm from those things.” (Clinic 5)* *“…Seeing how other agencies are understandably impacted by COVID—either not being able to provide telehealth services, or clinicians and staff being out on EDD, things like that. That’s also just been a huge adjustment…” (Clinic 7)* *“Even though tele-medicine and tele-video visits are available by the psychiatrist, I think it took them a few months to initially set it up. Probably with the insurances, with the reimbursements and so on. We couldn’t get access for at least from March to June and then we were able to get some access. Even with that. Now, they are completely booked out until October or so.” (Clinic 9)*
Subtheme: School-based Resources	*“Parents are really stressed with the younger kids and having to work full-time, be a teacher, manage kids who are having ADHD [attention deficit hyperactivity disorder] or behavioral issues, and not getting as much support from the schools for kids that have learning disabilities.” (Clinic 1)* *“The school systems, particularly in the high schools, created a wellness center within our high school districts. And that was gaining acceptance among our students, and they were definitely using it, definitely I would encourage students who either they themselves, their parents were resistant, or they were worried about finances, that this was a great resource. Obviously, with COVID, that disappeared—or, it didn’t completely disappear, but the ease of getting to it just was convoluted.” (Clinic 1)*
**Provider-Level Domain**	
**Theme: Impact of the Pandemic on Providers**	
Subtheme: Provider Stress	*“I also think some of the therapists are getting a little overwhelmed themselves, and some I know have cut back hours because they’re personally having trouble managing all of it.” (Clinic 1)* *“As clinicians, you have your own fear—I mean I’m just going to put it out there, I don’t want to catch it. I don’t want to die from COVID. And we don’t have PPE.” (Clinic 3)* *“So, you’ve got the clinician’s stress, you’ve got the family’s stress, you’ve got the child’s stress…We count ourselves fortunate that we’re still open. I know lot of other practices are not and so there is that stress. It’s a tough time. So I think asking PCPs to take on managing something for which we didn’t necessarily sign up for in medical school—it’s a delicate ask right now.” (Clinic 3)* *“I was going to say, my entire bandwidth is on surviving the next six months. That’s my entire bandwidth. I have nothing left.” (Clinic 3)* *“There’s no national unified message from anyone so we are the de facto trusted reference.” (Clinic 3)*
Subtheme: Adapting Services to the Pandemic	*“And a few that kind of have more involved depression and anxiety—I was doing some like—I would check up on them weekly because some of them couldn’t get in to see their therapist or their anxiety got worse being at home with their family, so I was doing a weekly telemed check-in. That worked until they got in to see their own therapist.” (Clinic 2)* *“I think all of us have noticed our that all of our well-checks are 5–10 min longer because we have to start with the impact of the pandemic on the families. So all the wellness pieces that we normally do are emphasized even more because we are trying to help people get through an unusual time and with all these restrictions. So, I think all of us have noticed that that has just added to what we were doing already.” (Clinic 3)* *“Before, if anyone had tested positive for COVID, we had an internal process where someone would call and offer direct counseling to the families, to the parents as well….I know I had a lot of patients who—mom or dad was very stressed with someone in the house who tested positive, or the patient tested positive and wanted someone to talk to. So we had a direct internal way to refer families right after they got their COVID diagnosis.” (Clinic 7)* *“We have been responsive on multiple levels, including improving our workflows to speed access to care; certainly improving access through remote services, providing both telephone- and video-based responses, so that people can maintain shelter-in-place or social distancing; and then more expanded programming as well…it’s kind of a mix of prevention and clinical services.” (Clinic 8)*
**Patient-Level Domain**	
**Theme: Changes in Mental Health Symptomatology**	
Subtheme: Anxiety & Depression	*“Their anxiety got worse being at home with their family.” (Clinic 2)* *“As this has gone along further without a clear end sight, I think it has brought up a lot more feelings of being detached from peers, loneliness, depression.” (Clinic 5)* *“I have seen a lot of anxiety increasing in kids due to living inside. Fears of going outside, but also just not being able to have outdoors activities. I have moms who will call me and tell me that “my child’s anxiety levels are getting worse.” Also, due to online classes, it’s affecting some of our families, our children; they’re struggling more with classes online. Just the anxiety of not knowing or not having what they need at home.” (Clinic 7)* *“We’ve done some surveying of community needs after COVID hit and shelter-in-place, and those surveys have, as would be expected, revealed a heightened level of anxiety and recurring trauma, depressive feelings, sense of chaos and shortage in the world, and just in general a heightened need.” (Clinic 8)* *“I’ve seen increased loneliness for kids. Especially the ones who don’t have close relationships with their parents or their family members.” (Clinic 9)*
Subtheme: Somatization	*“I feel like I’m seeing somatization if that’s the right word. I mean the number of kids in the past two and a half weeks who have had stomach aches or accidents, and those are probably the main two, but I mean just the last two and a half weeks is probably more than the previous three months put together. And it’s not 14-year-olds necessarily, it’s like 7-year-olds, 8-year-olds, 5-year-olds who I don’t think have the—to put it together to say that it’s the specter of COVID but it’s being at home, being at home, and being at home, and then still being at home and mom and dad are working, and not seeing their friends, and not going outside and playing.” (Clinic 3)* *“There is a lot more somatization but we’re pediatricians so I don’t know if your stomach[ache] is because of COVID or we’ve still had appendectomies, we still have kids diagnosed with brain tumors, we still have diabetes.” (Clinic 3)* *“We have seen patients coming in not only with anxiety and depression, but we are also seeing diabetics getting worse, blood pressure, hypertension’s getting worse; we have kids who otherwise are really healthy but exhibiting anxiety-like symptoms; noncompliance with medications, noncompliance with their usual health care, dietary noncompliance. So we’re seeing some effects of what we feel, or what we think, is more behavioral health-related, even in the medical aspect, just showing up on the blood work, and things like that.” (Clinic 8)*
Subtheme: Other Symptoms	*“I’m seeing a lot of video addiction, video gaming addiction, especially when there’s shelter in place” (Clinic 4)* *“Three year olds, four year olds, and five year olds, are really starting to act out and say kind of troubling things about wanting to die and wanting to kill themselves.” (Clinic 4)* *“I felt like we were seeing especially in the beginning of more intense cases like [unknown] psychotic breaks for some of our young adults, or [unknown] manic episodes, and suicide.” (Clinic 6)* *“I have about three patients—three to four patients a month running away from home. Because they were just—they just felt stuck… In the beginning I just didn’t know what to do. They run away and then they come back three or four days later. Or just walk away from home, take an Uber and go from home for a few hours or half a day then come back. That was something new…But just running away from home. That was something new I saw.” (Clinic 9)*
Subtheme: Symptomatic Improvement	*“They’re certainly not overwhelmed with having to stay up until 2am to finish their schoolwork, and they don’t have all those extracurriculars, so it has been an opportunity for people [to access mental health care].” (Clinic 1)* *“A lot of the kids who are suffering from some social anxiety because they were going to school, all of a sudden they felt great! It was really kind of sad to me that going to high school was more stressful than a pandemic… The anxiety levels were just dropping way down. Kids were so happy just being with their families… Initially these kids felt great, and now that social anxiety has really ballooned and the thought of going back to school, though they’re not going back to school, but the thought of it was kind of horrifying to them…” (Clinic 5)* *“…to cite one case and example—a 12-year-old girl with [disease] who’d had a surgical procedure and still had issues…and was being bullied for her obesity. Had been told she was a Chinese with Coronavirus even though she’s actually African American. She was just being bullied left and right including being kicked—physically injured on [the appendage] that she’d had surgery on. The school was not intervening…her life was just so peaceful after [shelter in place]. There were several like that because they deal with a lot of bullying and school anxiety. I do a lot of letter writing to teachers and principals regarding these issues that just don’t seem to be addressed adequately. So, I did see that improvement which was sort of sad like you said.” (Clinic 5)* *“I have not faced the level of anxiety or depression I expected with patients who are in social isolation or missing friends… I have not had to refer anybody, specifically, for significant enough anxiety or depression related to COVID anytime recently.” (Clinic 5)* *“I think there’s a subset of patients who are actually thriving in this pandemic, for I think the wrong reasons, not having to go to school, not having to constantly face the challenges that they were having academically.” (Clinic 7)*
Subtheme: Cascade of Stress from Parent to Child	*“I was talking with a friend this weekend about just a regular old 6-month check up that for the most part are generally straight forward are inescapable to go through without addressing the burden that is put upon the family—that is being put on families and then parents and then transferred to the kids. Not necessarily like the 6-month-old but the simple things. It’s just—I think it’s just massive.” (Clinic 3)* *“The stress that the family is going through certainly bleeds onto the children.” (Clinic 5)* *“And then I also have a subset of kids and families as a whole that are very directly impacted by this. I think in our country, we see a lot of disparity in the way that COVID is really impacting our community, and, to me, it has been surprising to see how much that is directly impacting the kids themselves… It has been a pretty rude awakening to see that these disparities do trickle down, even to young kids; even when they’re not discussed directly, they feel the limits some way or another. And that’s been pretty evident due to this pandemic.” (Clinic 7)*
Subtheme: Increased Utilization of Behavioral Health Resources	*“I’ve definitely seen more patients who… yeah, their stress levels went up, and they requested some kind of therapy, or their parents requested some kind of therapy directly related to their COVID diagnosis.” (Clinic 7)* *“We’ve seen a 25% increase in utilization of behavioral health services in our system across our three primary clinics.” (Clinic 8)*
**Theme: Social Determinants of Health**	
Subtheme: Socioeconomic Status	*“I feel like a lot of our families and their kids seem to be coping really well and that might be kind of our demographics. Where – it’s easier. They have houses with yards and some of them are still having help come in…it has been certainly challenging but not to the point where it’s devastating.” (Clinic 2)* *“I also think there are huge national events and huge international event like what happened in Minneapolis is adding another layer of trauma to what people are already struggling with in terms of economic uncertainty and COVID.” (Clinic 3)* *“For many families this is a medical crisis but even almost more pressing an economic crisis.” (Clinic 5)* *“We’re getting a lot of new referrals because people have lost insurance.” (Clinic 5)* *[Regarding barriers affecting the referral process] “Getting people on the phone. Not being able to pay their bill so then their phone number changes. Then they’re not coming in in-person and are they not answering the phone because they’re working, because their phone is not working anymore, because they lost it, because it’s broken?” (Clinic 6)* *“…kids worrying for their parents having to go out and work, experiencing severe anxiety around their parents working too much or coming home with the virus, and just a lot of it being that the community that we are in are forced to go to work…it adds these different layers of complexity when you work in a community that has deprivation in the services that are available to families.” (Clinic 7)* *“For those without homes, having a reliable communication technology and service [would help them engage in care].” (Clinic 8)*

## Data Availability

The data presented in this study are available on request from the corresponding author.

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
