# Peer review of "Pediatric Primary Care Perspectives of Mental Health Services Delivery during the COVID-19 Pandemic"

_children, 2022, doi:10.3390/children9081167_

Round 1

Reviewer 1 Report

The manuscript with the title ,,Pediatric Primary Care Perspectives of Mental Health Services 2 Delivery During the COVID-19 Pandemic” it is relatively well structured and tries to show what the needs of children and adolescents with mental health problems are at different times of the pandemic and how the medical staff intervenes in order to improve behavioral health programs.

There are 18 references, of which a number of 4 are over 10 years old, 4 references between 5-10 years old and 10 from the last 5 years.

There are no self-citations.

The manuscript involves the collection of data from 9 US health care providers for a period of 5 months in 2020. There is also approval of the ethics committee. The study was conducted online, the discussions between the researcher and the patient were recorded and lasted about 45 minutes. The demographic situation of the study participants was also analyzed. The study began with the onset of the COVID-19 pandemic in the United States, so additional questions were added to the basics.

It is interesting that 5 main topics were identified, including the impact of the pandemic on service providers, the limited availability of behavioral health resources, the possibility of rapid transition to virtual meetings. From here, 3 directions in the study followed: at system level, the medical service provider and the patient.

During the pandemic, there were changes in mental health symptoms. Another aspect approached in the study is the one related to the difficulties faced by the families of the patients in the study during the pandemic. The study seeks to verify the hypothesis that the pandemic caused changes in the mental health of children and adolescents, but also increased patient symptoms and socio-economic stress.

The study is actually an assessment of a situation, the data obtained can be used later in research.

 The conclusions are in line with those presented by the authors in the manuscript.

The data in the tables are presented in detail and are easy to interpret.

Lines 56-58: ,, over one-third (34.8%) of 56 youth between the ages of 2-21 received their outpatient mental health care from a PCP 57 only.[9]”.

How did you apply the interview to the children online? Especially the 2 year olds? Can you detail?

Lines 155-158 : ,, However,  loss of the patient-provider connection was a frequent concern as providers reported challenges engaging with patients, especially young children, and building relationships with  new patients via telehealth.”.

 How could the patient-provider connection be achieved, especially for young children and through telemedicine

Lines 205-211 : ,, Some examples included  shifting clinic services to remote formats, having weekly telehealth check-ins with patients  who were unable to see their therapist, and creating internal processes to offer direct counseling to households in which a family member tested positive for COVID-19. Another  stated impact on providers was the need for more time to interact with patients during  check-ups, video visits, and phone calls to address the impact of COVID-19 on patients’  lives and health.”.

Can you detail? Have health services changed or just adapted to the new requirements? In terms of time, has it exceeded the 45 minutes previously expected?

Lines 232-237: ,, Conversely, many providers also reported that school closures during the pandemic  had positive effects on mental health for some of their patients. They observed that students affected by social anxiety, bullying, and academic challenges experienced relief  from school-based stressors while sheltering-in-place: “A lot of the kids who are suffering  from some social anxiety because they were going to school, all of a sudden they felt great” (Clinic  5) 

Can you detail? Are there studies regarding this?

Lines 271-276: ,, While there was some variation in experience, many providers  reported perceiving a reduction in available behavioral health resources. Providers reported long wait times for patient mental health referrals that existed well before the pandemic. The significant challenge in referring patients to specialty mental health care due  to the limited mental health workforce could be mitigated by integrated and consultative  care models “

Can you give more details? Do you refer here to the aspect regarding the human resource and the addressability time of the patients? Can you mention studies in this regard?

 Lines  284-288 :Providers described how patients’ pandemic living situations (e.g. overcrowding, lack of private spaces) affected mental health. Providers also discussed elevated stress among families with essential workers due to increased risk for infection or  loss of employment. The provision of mental health care requires awareness of patients’  SES and resources both during and after the COVID-19 pandemic.

Can you comment? Do you consider that it is necessary to open private spaces to provide medical care, especially during the pandemic, to patients with these diseases?

Author Response

Reviewer #1:

  1. There are 18 references, of which a number of 4 are over 10 years old, 4 references between 5-10 years old and 10 from the last 5 years.

Response: We added 7 new references from the last 2 years.

  1. Lines 56-58: ,, over one-third (34.8%) of 56 youth between the ages of 2-21 received their outpatient mental health care from a PCP 57 only.[9]”.

How did you apply the interview to the children online? Especially the 2 year olds? Can you detail?

Response: Thank you for this question. The interview participants in our study were all pediatric primary care clinicians who work with youth, and not pediatric patients. We did not interview any minors or youth in our study.

  1. Lines 155-158 : ,, However,  loss of the patient-provider connection was a frequent concern as providers reported challenges engaging with patients, especially young children, and building relationships with  new patients via telehealth.”.

How could the patient-provider connection be achieved, especially for young children and through telemedicine?

Response: Thank you for this feedback. We added a comment and citations in the discussion to address this: “Providers also acknowledged difficulty in engaging with pediatric patients, particularly young children, via telehealth, which has been cited in the literature as a significant barrier to remote mental health treatment.[18, 19] However, clinicians may work on engaging the adult caregiver as an ally in establishing a therapeutic alliance with the child.

  1. Lines 205-211 : ,, Some examples included  shifting clinic services to remote formats, having weekly telehealth check-ins with patients  who were unable to see their therapist, and creating internal processes to offer direct counseling to households in which a family member tested positive for COVID-19. Another  stated impact on providers was the need for more time to interact with patients during  check-ups, video visits, and phone calls to address the impact of COVID-19 on patients’  lives and health.”.

Can you detail? Have health services changed or just adapted to the new requirements? In terms of time, has it exceeded the 45 minutes previously expected?

Response: Thank you for this comment. We have clarified in the discussion: “Overall, healthcare services changed fundamentally at the system level.” Additionally, we added to the discussion: “Clinicians noted that there was the need for more time to interact with patients during telehealth encounters to address the impact COVID-19 on patients’ lives. Because the focus groups were conducted early in the pandemic, this finding may be attributable to an initial adjustment to the pandemic.” Unfortunately, we do not have data quantifying the amount of time required.

  1. Lines 232-237: ,, Conversely, many providers also reported that school closures during the pandemic  had positive effects on mental health for some of their patients. They observed that students affected by social anxiety, bullying, and academic challenges experienced relief  from school-based stressors while sheltering-in-place: “A lot of the kids who are suffering  from some social anxiety because they were going to school, all of a sudden they felt great” (Clinic  5) 

Can you detail? Are there studies regarding this?

Response: Thank you for this question. We added some detail and cited a study in the discussion: “While some providers in our study noticed that children with social anxiety, bullying, and academic challenges initially experienced relief during school closures, the literature supports that lockdown conditions prevented exposure to feared situations and reinforced avoidant behaviors, likely making existing mental health conditions worse when schools reopened.[22]

  1. Lines 271-276: ,, While there was some variation in experience, many providers  reported perceiving a reduction in available behavioral health resources. Providers reported long wait times for patient mental health referrals that existed well before the pandemic. The significant challenge in referring patients to specialty mental health care due  to the limited mental health workforce could be mitigated by integrated and consultative  care models “

Can you give more details? Do you refer here to the aspect regarding the human resource and the addressability time of the patients? Can you mention studies in this regard?

Response: Thank you for this suggestion. We added a clarification with citations: “The significant challenge in referring patients to specialty mental health care due to the limited mental health workforce could be mitigated by integrated and consultative care models, such as the Collaborative Care Model or Child Psychiatry Access Programs, which help to extend the reach of child mental health specialists.[21, 22]”

  1. Lines  284-288 :Providers described how patients’ pandemic living situations (e.g. overcrowding, lack of private spaces) affected mental health. Providers also discussed elevated stress among families with essential workers due to increased risk for infection or  loss of employment. The provision of mental health care requires awareness of patients’  SES and resources both during and after the COVID-19 pandemic.

Can you comment? Do you consider that it is necessary to open private spaces to provide medical care, especially during the pandemic, to patients with these diseases?

Response: We appreciate this question, and we have added additional content in the discussion: “Other clinical literature has noted the financial burden of telehealth on families,[18] and this speaks to the need to partner with government or community agencies that may provide technology resources. If a PCP is sensing that a patient is lacking privacy at home, then that patient may need to be prioritized for an in-person visit.

Reviewer 2 Report

General Comments

This article seeks to explore the experiences of pediatric PCPs and clinical staff providing mental health care during the pandemic, in order to clarify the mental health needs of children and adolescents during the COVID-19 pandemic, as well as the increasing complexity of mental health problems that PCPs face in primary care.

Introduction:

The theoretical framework seems adequate although quite succinct, perhaps an approach on the factors that can influence the children's mental health needs as well as the factors that influence the quality of care provided, could be clarifying.

Materials and Methods:

The methods used, the instruments, sample recruitment, time horizon of the study, the resources used are explained. As well as safeguarding ethical and legal issues.

Results:

The result comes essentially from the information collected from doctors. At the level of primary health care, it would be important to collect the opinion of other health professionals.

The presentation of the data is clear, table 1 and fig 1 help to clarify the data where the areas and themes where the pandemic impacted the provision of care are clearly identified.

Discussion:

The results are in line with those of other researchers where it is recognized that the covid 19 pandemic had a very important impact on the provision of health care.

The limitations of the study are acknowledged.

Conclusion.

The conclusions are not very significant, adding little to the knowledge that already exists on the necessary approximation of specialty mental health and primary care.

Author Response

Reviewer #2:

  1. Introduction: The theoretical framework seems adequate although quite succinct, perhaps an approach on the factors that can influence the children's mental health needs as well as the factors that influence the quality of care provided, could be clarifying.

Response: Thank you for this suggestion. We have added an additional sentence to clarify: “We chose this framework because the social-ecological model recognizes that an individual child does not exist in isolation but is nested within family and clinician relationships, community contexts, and social policies.”

  1. Results: The theoretical framework seems adequate although quite succinct, perhaps an approach on the factors that can influence the children's mental health needs as well as the factors that influence the quality of care provided, could be clarifying.

Response: Thank you for this comment. Please see the response to #1 above.

  1. Conclusion: The conclusions are not very significant, adding little to the knowledge that already exists on the necessary approximation of specialty mental health and primary care.

Response: We appreciate the reviewer’s feedback. We have added revised our discussion and conclusion to emphasize the need to support primary care provider well-being and work experience as a key lesson from the stresses of the COVID-19 pandemic.
